# Patient health records and whole viral genomes from an early SARS-CoV-2 outbreak in a Quebec hospital reveal features associated with favorable outcomes

**Bastien Paré[1,2]☯, Marieke Rozendaal[1]☯, Sacha Morin[3,4], Léa Kaufmann[1,2], Shawn M. Simpson[1], Raphaël Poujol[5], Fatima Mostefai[2,5], Jean-Christophe Grenier[5], Henry Xing[1], Miguelle Sanchez[6], Ariane Yechouron[6], Ronald Racette[7], Julie G. Hussin[5,8], Guy Wolf[4,9], Ivan Pavlov[7]\*, Martin A. Smith[1,2]\***

1 CHU Sainte-Justine Research Centre, Montreal, Quebec, Canada, 2 Department of Biochemistry and Molecular Medicine, Faculty of Medicine, Université de Montreal, Quebec, Canada, 3 Department of Computer Science and Operational Research, Université de Montréal, Montreal, Quebec, Canada, 4 Mila—Quebec AI Institute, Montreal, Quebec, Canada, 5 Montreal Heart Institute, Montréal, Quebec, Canada, 6 Department of Microbiology and Infectious Diseases, Hôpital de Verdun, Montreal, Quebec, Canada, 7 Department of Emergency Medicine, Hôpital de Verdun, Montreal, Quebec, Canada, 8 Department of Medicine, Faculty of Medicine, Université de Montréal, Montreal, Quebec, Canada, 9 Department of Mathematics and Statistics, Université de Montréal, Montreal, Quebec, Canada

☯ These authors contributed equally to this work.
\* martinalexandersmith@gmail.com (MAS); ivan.pavlov.md@gmail.com (IP)

**Data Availability Statement:** Basecalled, demultiplexed and size-filtered reads can be found via SRA under bioproject PRJNA730334. Scripts,

## Abstract

The first confirmed case of COVID-19 in Quebec, Canada, occurred at Verdun Hospital on February 25, 2020. A month later, a localized outbreak was observed at this hospital. We performed tiled amplicon whole genome nanopore sequencing on nasopharyngeal swabs from all SARS-CoV-2 positive samples from 31 March to 17 April 2020 in 2 local hospitals to assess viral diversity (unknown at the time in Quebec) and potential associations with clinical outcomes. We report 264 viral genomes from 242 individuals–both staff and patients–with associated clinical features and outcomes, as well as longitudinal samples and technical replicates. Viral lineage assessment identified multiple subclades in both hospitals, with a predominant subclade in the Verdun outbreak, indicative of hospital-acquired transmission. Dimensionality reduction identified two subclades with mutations of clinical interest, namely in the Spike protein, that evaded supervised lineage assignment methods–including Pangolin and NextClade supervised lineage assignment tools. We also report that certain symptoms (headache, myalgia and sore throat) are significantly associated with favorable patient outcomes. Our findings demonstrate the strength of unsupervised, data-driven analyses whilst suggesting that caution should be used when employing supervised genomic workflows, particularly during the early stages of a pandemic.

statistical analyses, consensus genomes and variant files can be found at: https://github.com/TheRealSmithLab/Verdun.

**Funding:** SM is supported by an IVADO MSc excellence scholarship and an FRQNT B1X scholarship. JGH is a Fonds de Reherche du Québec en Santé Research Scholar (252997) funded by IVADO COVID19 Rapid Response grant (CVD19-030) and the Montreal Heart Institute Foundation. GW is supported by Canada CIFAR AI Chair. MAS is supported by a Fonds de Reherche du Québec en Santé Junior 1 fellowship (295760).

**Competing interests:** IP has received speaking fees and an unrestricted research grant from Fisher & Paykel Healthcare Limited (Auckland, New Zealand) and an unrestricted research grant from OpenAI inc (San Francisco, CA, USA). MAS has received research consumables, travel and accommodation expenses to speak at Oxford Nanopore Technologies conferences. Otherwise, the authors declare that the submitted work was carried out in the absence of any professional or financial relationships that could potentially be construed as a conflict of interest.

## Introduction

The first confirmed case of COVID-19 in the province of Quebec, Canada was seen at Verdun Hospital, a 244-bed general adult hospital in Montreal, on February 25, 2020. Community transmission was confirmed in the following weeks, and culminated in a localized outbreak in hospitalized patients. On March 30th, a hospital-wide screening of all admitted asymptomatic patients was performed, 45.2% of whom had detectable levels of SARS-CoV-2 RNA. A policy of universal testing before hospital admission was rapidly established, but did not prevent further smaller localized outbreaks. At the time, there was no publicly available information on SARS-CoV-2 lineage diversity in Quebec. Moreover, reports of asymptomatic and presymptomatic infection and transmission were only beginning to emerge [1–3]. It has since been established that infected individuals display a range of symptoms of variable clinical severity [4–7].

In addition to its epidemiological utility, global viral genome sequencing efforts have revealed that SARS-CoV-2 has (and will continue to) quickly evolved, diversified and adapted to the selective pressures of a new mammalian host and large-scale vaccination efforts [8]. The accessibility and affordability of Oxford Nanopore sequencing has facilitated the global adoption of genomic epidemiology during the pandemic. Although once stigmatizing, the error-rate of nanopore sequencing has significantly dropped in the last few years [9, 10], facilitating the reliable generation of consensus sequences–particularly when coupled to expert-backed community-developed analytical pipelines, such as the one rapidly disseminated by the ARTIC Network [11]. The resulting genome consensus is then used for multiple sequence alignment and phylogenetic analysis. Comparing hundreds and thousands of viral genomes and their genetic variants can be a daunting and computationally expensive task. Therefore, genotyping or lineage assignment tools are often used instead of unsupervised approaches for the classification of genomes by leveraging previously-generated phylogenies or lists of curated signature mutations to associate a genome with a pre-defined clade. The most popular lineage assignment tools are Pangolin [12], which uses a supervised learning approach to classify sequences, and Nextstrain/Nextclade [13], which use condensed phylogeny and phylogenetic placement methods, respectively.

An alternative computational strategy for the visualization and interpretation of high-dimensional data, such as that queried when comparing hundreds of viral genomes, is dimensionality reduction. These unsupervised machine learning methods decompose large datasets by projecting dependencies and relationships between data into lower dimensional space (usually 2D). Principal component analysis (PCA), uniform manifold approximation and projection (UMAP) and t-distributed stochastic neighbor embedding (tSNE) are popular dimensionality reduction algorithms commonly used in bioinformatics and genomics [14–16]. Potential of heat diffusion for affinity-based transition embedding (PHATE) is a recently developed dimensionality reduction method that was shown to outperform all major dimensionality reduction algorithms at denoising and preserving desirable properties of the surveyed data (Moon et al. 2019). PHATE can be used to extract clusters of data points and as a template to calculate the sample-associated density estimate and relative likelihood, as enabled by the MELD method [17]. Unsupervised learning algorithms can therefore be an efficient alternative to phylogeny for genomic epidemiology, amongst other applications.

In this study, we interrogated the viral genomic diversity and patient outcomes of 242 SARS-CoV-2 infections in a local, first-wave outbreak. We describe the clinical features of this cohort, including symptoms and mortality, and present longitudinal sequencing data from 21 infected individuals. We also compare popular lineage assignment tools with phylogenetic and

dimensionality reduction techniques to interrogate the link between viral genomic diversity, symptomology and clinical outcomes.

# Results

## Clinical observations and outcomes

We retrieved the nasopharyngeal swabs and medical records for 242 individuals (267 samples) who tested positive for the presence of viral nucleic acids between 31 March and 17 April 2020 at Verdun and Notre-Dame hospitals in Montreal, Canada. Table 1 presents general patient data that were included in this study. 163 individuals that participated in this study were female, 79 were male and their age spanned from 2 to 104 years old, with the median age being 50. About half of the participants (134) were hospital employees. 223 individuals presented COVID-19 symptoms before receiving a positive SARS-CoV-2 diagnosis, 21 patients developed symptoms after diagnosis (presymptomatic) and 16 remained asymptomatic. For most patients, symptoms associated with the SARS-CoV-19 infection were the presence of fever, coughing and dyspnea. 87 individuals required hospitalization, of which 23 died during admission. Logistic regression of generic patient data identified age (P>|z| = 0.05520) and the presence of comorbidities (Charlson index >0; P>|z| = 0.00238) over sex, hospital and employee status as the main covariates predictive of mortality, as expected. The Cycle Number (CN) score–a diagnostic measure of viral load from nasopharyngeal samples akin to the cycle threshold (Ct) used in quantitative PCR–was not significantly associated with comorbidities (Wilcoxon rank sum test). However, we observed a slight yet significant positive correlation (0.30, Pearson's product-moment correlation) between CN and the Charlson comorbidity index. The Charlson comorbidity index is a quantitative metric premised on clinical features and

**Table 1. Cohort summary.**

|  |  | n | % |
|---|---|---|---|
| **Individuals** | *Total* | *242* |  |
|  | Male | 79 | 32.6 |
|  | Female | 163 | 66.9 |
|  | Employee | 134 | 55.4 |
|  | Hospitalized | 87 | 36 |
| **Clinical presentation** | Symptomatic | 205 | 84.7 |
|  | Presymptomatic | 21 | 8.7 |
|  | Asymptomatic | 16 | 6.6 |
| **Reason for hospitalization** | COVID-19 | 54 | 62.1 |
|  | Other | 33 | 37.9 |
| **Patient outcome** | Deceased | 23 | 9.5 |
|  | Survival or unknown outcome | 219 | 90.5 |
| **Age** | 0–30 | 27 | 11.2 |
|  | 31–60 | 134 | 55.3 |
|  | 61–104 | 81 | 33.5 |
| **Charlson comorbidity index** | 0* | 187 | 77.3 |
|  | 1 | 7 | 2.9 |
|  | 2 | 26 | 10.7 |
|  | 3 | 16 | 6.6 |
|  | 4+ | 6 | 2.5 |

* Includes individuals with no reported comorbidities.

developed to predict the ten-year mortality for a patient who may have a range of comorbid conditions.

## Viral RNA abundance and bioinformatics parameters affect genome assembly quality

Of the 267 samples, RNA was extracted from 264 samples (240 individuals) and subjected to tiled amplicon sequencing (see Methods), generating a median reference genome coverage (i.e. completeness) of 97.7% from a median of 298,817 filtered reads per sample, including 70 full-length (99.6% complete) genomes (Fig 1 and S1 Table). Of the 12 negative controls, only the sample generated using version 1 of the ARTIC Network protocol generated a false positive genome that could be assigned to a SARS-CoV-2 subclade (S1 Fig, S1 and S2 Tables). A subset of samples (24) with low read coverage was re-sequenced to improve genome completeness and act as a technical replicate (S3 Fig and S4 Table), which recovered 3 samples that were below 80% completeness and another 3 under 90%. Of the 264 sequenced genomes, 207 with at least 90% genome completeness were uploaded to GISAID [18] as soon as the genomes were assembled, including the first publicly disseminated SARS-CoV-2 genomes from Quebec (c.f. submitting laboratory: Smith Laboratory, Centre de Recherche CHU Sainte-Justine).

There is a clear, expected correlation between genome completeness and viral RNA abundance, as measured via the CN score generated during diagnosis (Fig 1B). As the number of PCR cycles used was based on RNA abundance as indicated by the CN score (see Methods) a similar trend in final genome completeness is also observed for this metric. Obtaining full genomes from samples with high CN values required more depth of sequencing given disparities in amplicon coverage, which are exacerbated in samples with lower input RNA. An overview of all amplicons, their abundances and associated negative controls are displayed in S1 Fig. A final set of 237 genomes with at least 80% genome completeness was retained for subsequent analyses.

We assessed the impact of two different versions of the ARTIC Network bioinformatics pipeline on consensus genome production: (i) The default version using signal-level correction with Nanopolish and (ii) the experimental version, which uses the Medaka neural network and Longshot variant caller. Excluding regions covered by less than 20 reads, the Nanopolish version generated an average of $2.9 \pm 3.5$ (standard deviation) ambiguous bases ('N') per genome, versus $1.1 \pm 2.5$ for Medaka, suggesting that the experimental version performs better than the default parameters. Closer inspection of the ambiguous bases in the consensus sequences revealed that these positions were broadly associated with variant allele frequencies (VAF) below ~0.9. We noticed that many of these variant positions were important for SARS-CoV-2 subclade assignment, therefore we replaced ambiguous bases in the consensus sequence with the most dominant variant (for VAF >0.5 only). Interestingly, 5 genomes presented mean VAF (mVAF) scores below 0.9 despite having CN scores <15 and 25 cycles of PCR, suggesting that more than one viral genome haplotype may be present (Fig 2).

Of note, 42/237 genomes with ≥80% completeness had an incomplete S gene and 90/237 had an incomplete N gene. The latter harbors one of the consistently less abundant amplicons from the ARTIC V3 PCR amplification scheme (S1 Fig). However, only 4 unique mutations (7 mutations in total) were observed in 147 genomes with complete N genes, suggesting that the missing sequences are unlikely to contain many mutations. Few mutations were also observed for the S gene; besides the D614G mutation (present in all but one genome), 9 genomes had an A24782G mutation (N1074D substitution) and 7 had a G21641T (A27S substitution).

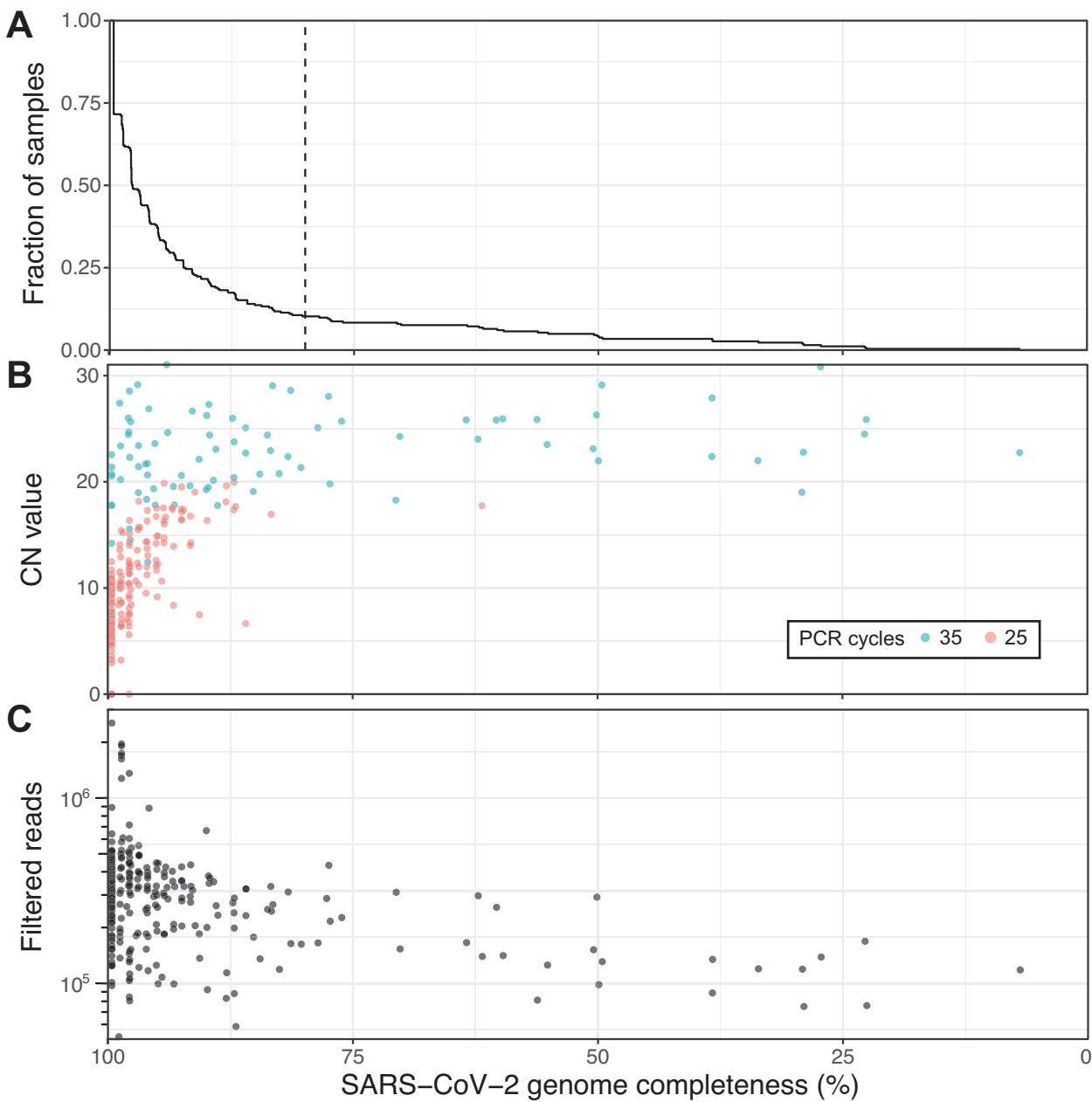

**Fig 1. Viral genome sequencing of 264 SARS-CoV-2 samples with Oxford nanopore.** (A) Cumulative distribution of genome completeness using the ARTIC bioinformatics SOP (see methods). Dashed vertical line corresponds to the 80% completeness threshold used for phylogenetic reconstruction. (B) Relationship between CN score at diagnosis, as measured by the Abbott RealTime M2000rt device (higher CN = lower viral load), and genome completeness. (C) Relationship between number of quality passed reads filtered using the ARTIC bioinformatics SOP and genome completeness.

## Unsupervised machine learning outperforms supervised methods at discriminating between viral subclades

The first batch of 5 SARS-CoV-2 genomes were generated 3 days after receiving all the samples and uploaded to GISAID shortly thereafter. From these first genomes, a preliminary phylogenetic analysis revealed that at least 2 different subclades were present in the Verdun hospital

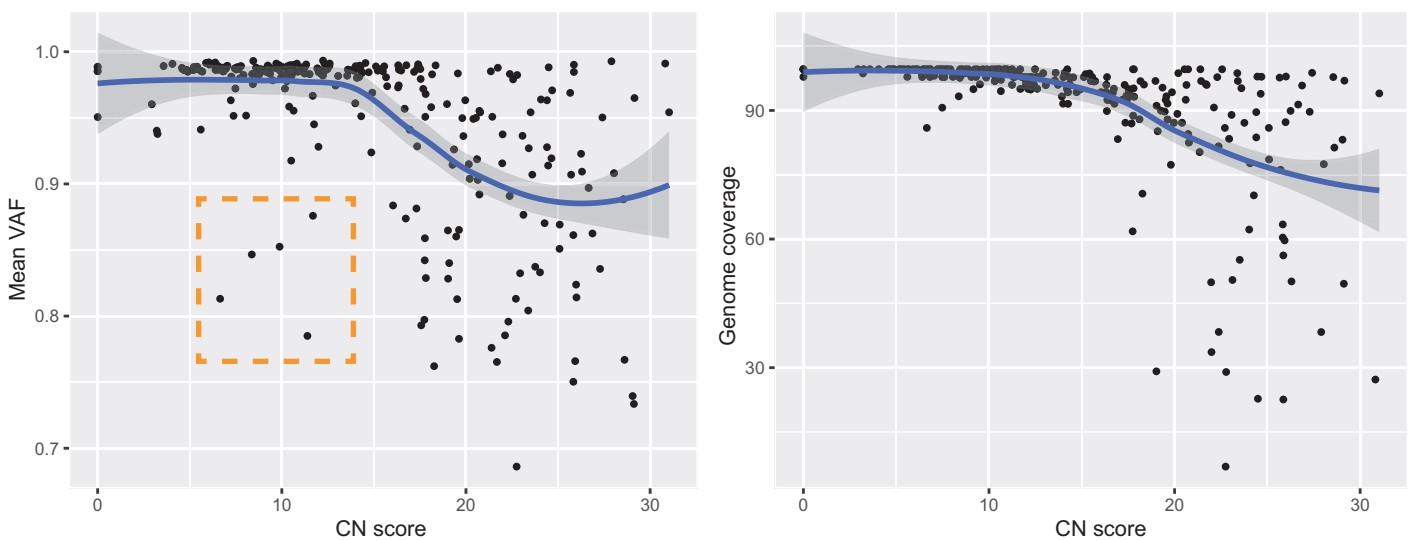

**Fig 2. Genomic features in relation to RNA abundance at diagnosis.** Average variant allele frequency (VAF) in function of RNA abundance (left). Genome completeness in function of RNA abundance (right). Outliers highlighted in orange. CN = Cycle Number.

outbreak (not shown). The full phylogenetic relationship of all SARS-CoV-2 genomes with 80% or more completeness is displayed in Fig 3, exposing the presence of a dominant subclade, consistent with suspected nosocomial infection. SARS-CoV-2 lineage classification with Pangolin [12] confirmed that the main cluster mainly consists of subclade B.1, which represents 65.17% (174) of the classified genomes. Other common subclades included B.1.147 (35) and B (35), while a mix of other B lineages was predicted for the remaining samples. NextClade lineage assignment was more conservative, emitting classifications for 205 genomes: 7x 19A, 169x 20A and 29x 20C (S1 Table). When compared to viral diversity from subsequent infections world-wide, only genomes classified as subclade 20C in our samples (corresponding to a subset of the pangolin B.1 classifications) may have persisted and diversified in the human population (S2 Fig).

However, discrepancies were observed between the phylogenetic analysis and Pangolin/ NextClade lineage classification, prompting us to employ an alternative genotyping strategy by grouping viral haplotypes based on the 22 most common (allele frequency above 10%) SARS-CoV-2 variants observed in the global community between December 2019 and July 2020, as reported in GISAID on January 20th 2021 (see Methods). The resulting haplotype groups were more consistent with the phylogenetic analysis than the Pangolin/NextClade classifications, prompting us to retain this genotyping strategy in subsequent analyses. A full list of the samples, their assigned subclades, and associated clinical features is listed in S1 Table while the genomic variants used for haplotype assignment are listed in S6 Table. No statistically significant correlation between haplotype assignments and clinical features was observed (Fisher's exact test, S6 Table).

As an alternative to phylogeny, we also applied PHATE [22] to visualize genomic variation and clinical features across all samples (see Fig 4 and Methods). PHATE relies on diffusion geometry to perform nonlinear dimensionality reduction of the data. The resulting representation preserves both local and long-range pairwise similarities, thereby offering a useful way to study how target variables are distributed across the genomic and clinical manifolds.

The genomic PHATE embeddings clearly delineate two subclades of haplotype group II, which are supported by the phylogenetic analysis but not by Pangolin lineage assignment

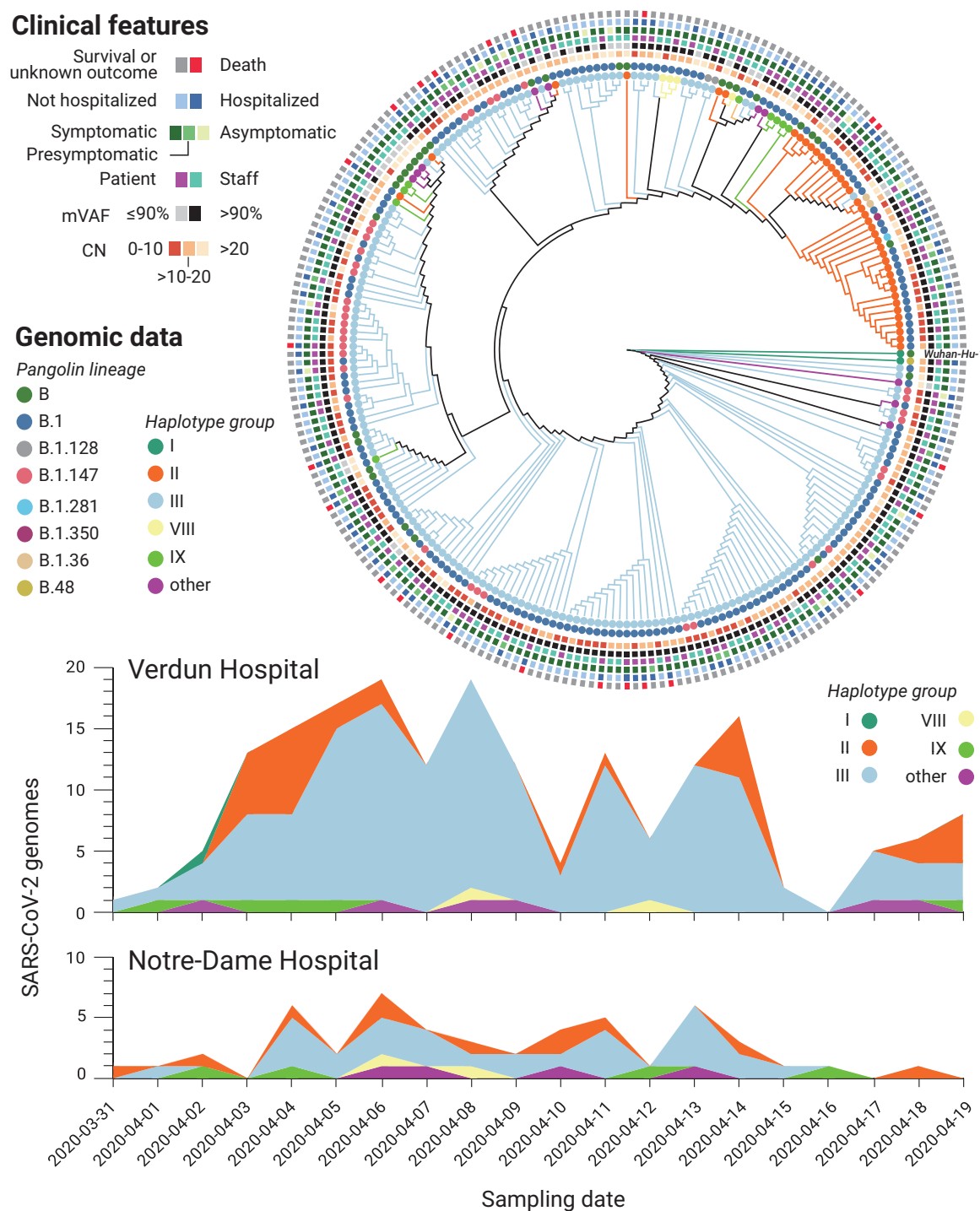

**Fig 3. Genomic and clinical features of 234 SARS-CoV-2 infections.** (Top) Maximum likelihood phylogenetic tree reconstruction of de novo assembled genomes spanning at least 80% of the SARS-CoV-2 reference genome (Wuhan-Hu-1) covered by ≥20 reads and annotated with clinical features of interest. The phylogeny was calculated from a multiple sequence alignment generated with MAFFT [19] using MEGA [20] and visualised with Iroki [21]. Lineage classification performed with Pangolin 2.1.10 [12] (outer circles) and haplotype assignment was performed based on the 20 most common variants in GISAID [18] from the first wave of the pandemic (inner circles). (Bottom) Haplotype diversity over time across two local hospitals. mVAF: Median variant allele frequency, CN: Cycle number.

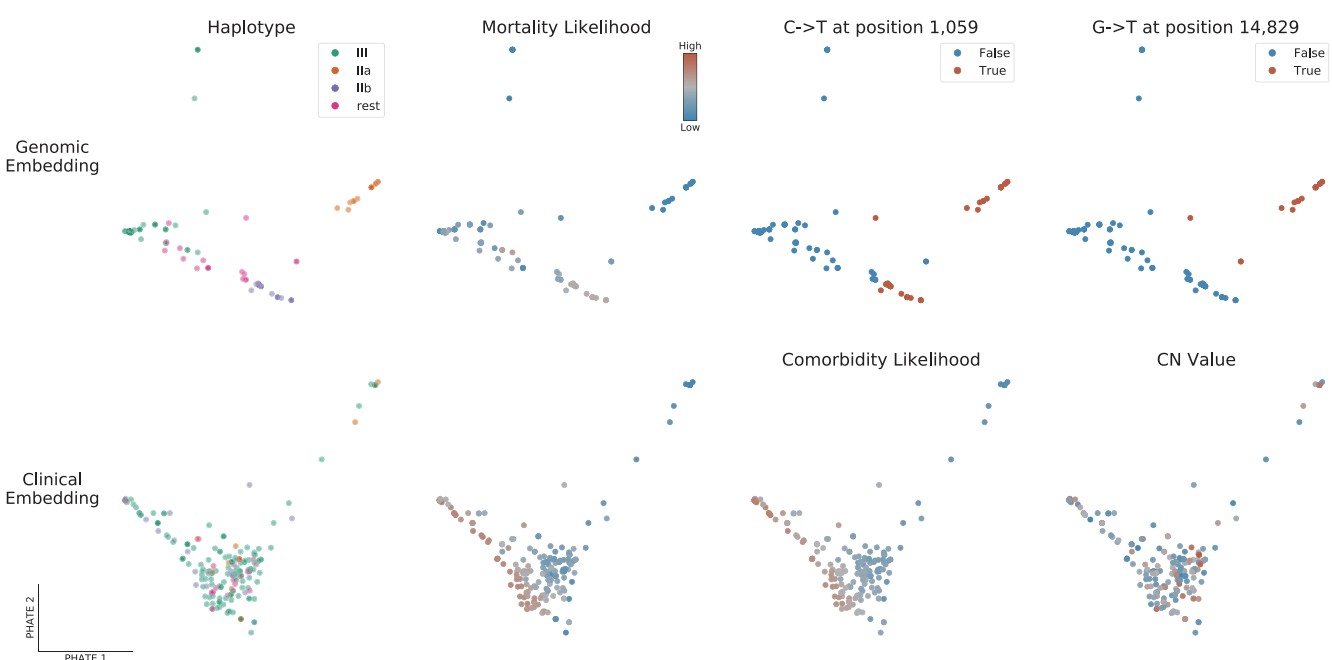

**Fig 4. PHATE embeddings of genomic and clinical features.** Two-dimensional PHATE embeddings of the genomes (Top) and of the clinical features (Bottom). Each marker represents one patient and the embedding location of a given patient indicates feature similarity with surrounding samples as well as dissimilarity with distant ones. The embeddings are unsupervised and the labels of interest are used for coloring only. Specifically, mortality and comorbidity likelihoods were computed using MELD [23], a graph signal processing tool used to smooth a binary variable on the patient-patient graph to determine which regions of its underlying data manifold are enriched or depleted in patients with a specific outcome.

(almost all genomes classified as B.1) or NextClade (all 20C). Subclade IIa (16 genomes) likely evolved from IIb (24 genomes) as its members share the same mutations plus 4 additional mutations, 3 of which are non-synonymous: 1150C>T, ORF1a.G295 (synonymous); 4886C>T, ORF1a.P1541S; 14829G>T, ORF1b.M454I; 27964C>T, ORF8.S24L). Interestingly, segregated non-synonymous S gene mutations were observed in both haplogroup IIa/b sub-clades; 24782A>G (S:N1074D) in 9/16 genomes from IIa and 21641G>T (S: A27S) in 5/24 genomes from IIb. The latter was also present in 2/176 haplogroup III genomes, while few other mutations were observed in the S gene, excluding the T23403G (S:D614G) mutation present in all but one genome. All 9 A24782G mutations in the N gene (see above) were uniquely present in haplogroup IIa genomes, whereas 5/7 G21641T mutations in the S gene were present in haplogroup IIb genomes (the remaining 2 in haplogroup III).

### Dimensionality reduction reveals associations between viral subclades, clinical features and patient outcomes

We next queried if there was a link between the observed viral heterogeneity and clinical features. To explore the high-dimensional data, we privileged a dimensionality reduction approach by applying MELD [23] to compute a likelihood gradient across the genomic PHATE embeddings (S4 and S5 Figs). We observed that the haplogroup IIa subclade appeared to be preferentially associated with certain clinical features, prompting us to divide the SARS-CoV-2 genomes into the 4 most dominant haplogroups (IIa, IIb, III and 'rest'). The PHATE-derived genotype groupings identified that headaches (p-value = 0.01773, Fisher's exact test), the presence of comorbidities (P = 0.01858) and, to a lesser extent, nausea, vomiting and

diarrhea (P = 0.0692) were linked to viral genotypes (S6 Table). The only symptom that was significantly associated with a specific S gene mutation was headaches (mutation S:N1074D, P = 0.01049). Interestingly, patients infected with haplogroup III were significantly more likely to present comorbidities (43/156) than other haplogroups (6/61, P = 0.006173).

Alongside viral genomics, we used PHATE to classify patient samples based on the diverse clinical features surveyed in this cohort (Fig 4, lower panel and S6 and S7 Figs). Coughing and fever are uniformly distributed across the cohort, with the exception of patients requiring breathing assistance for the former. However, patients along the lower-left quadrant are more likely to present severe clinical features (mortality, comorbidities, hospitalization, breathing assistance, etc) whereas those in the upper right were more likely to be employees or present flu-like symptoms (headache, myalgia, sore throat), suggesting that these symptoms are associated with favorable outcomes. Indeed, patients presenting these symptoms were ~10x less likely to die than those that didn't (odds-ratio 0.0951, P = $9.55 \times 10^{-5}$, Fisher's exact test). These results indicate that clinical features are more robust indicators of health outcomes than viral genotypes in this patient cohort [24–26].

## Comparative sampling reveals hospital-acquired transmission of SARS-CoV-2

To validate if the abundance of a particular subclade was due to hospital-acquired transmission or if the observed lineage frequencies were representative of SARS-CoV-2 lineages in the local community, we compared the distribution of viral genotypes between 2 different hospitals (Fig 3, bottom). The nearby Notre-Dame Hospital–which had no reported outbreaks of COVID-19 during the study period–had 29 of 58 (50.0%) infections attributed to haplogroup III, whereas this subclade was assigned to 146 out of 209 infections (69.9%) at Verdun Hospital, consistent with nosocomial transmission.

## Longitudinal sequencing of SARS-CoV-2 positive subjects

Among this cohort, 21 individuals (mostly hospital employees) were sampled more than once, thus enabling longitudinal analysis (S1 and S7 Tables). Of these, the viral genomes from 13 individuals were assigned a haplotype group for all time points. For all but 3 individuals, the viral load (inversely proportional to the CN value) decreased, while the median genome completeness decreased by 3.4%. As expected, the majority (24/37) of the genomes were composed of haplotype group III. Interestingly, we noticed that some patients presented different viral haplotypes depending on the sampling date (Fig 5). Although most of these occurrences are linked to very high (>25) CN scores–particularly the subsequent samples–some individuals presented more than one viral haplotype with respectable CN scores (individuals 93,100,152,159,205). Only 3 individuals had multiple samples with CN scores below 20, one of which (individual 78, a staff member) produced 2 haplotype III genomes with remarkably lower mVAF values, suggesting possible infection by more than one SARS-CoV-2 subclade.

## Discussion

This study provides a description of 264 SARS-CoV-2 genomes associated with 242 infections during one of the first reported local outbreaks in Canada. Quebec is among the provinces most affected by COVID-19 in Canada, with more than 360,000 cases and over 127,000 being confirmed on the island of Montreal alone (as of 2021-05-12). Using nanopore sequencing, we were able to resolve the genetic diversity of SARS-CoV-2 and confirm the presence of multiple subclades of the virus, as well as a dominant subclade indicative of hospital-acquired transmission at the Verdun hospital.

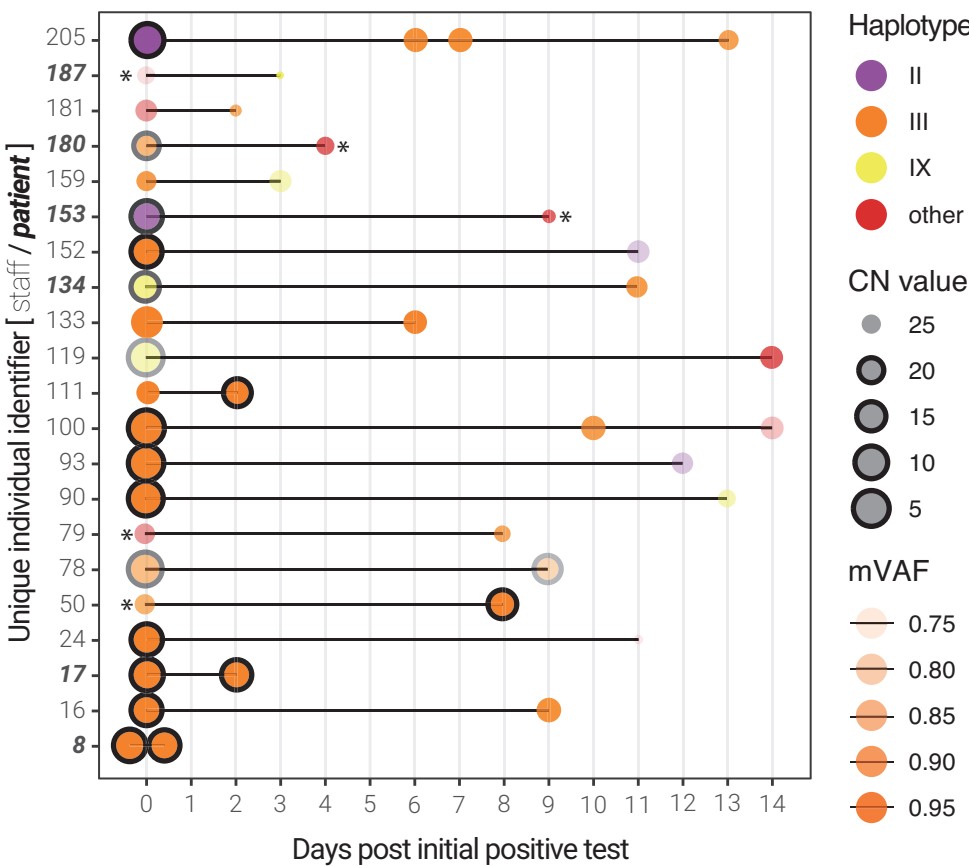

**Fig 5. Longitudinal sequencing of SARS-CoV-2 positive subjects.** Multiple samples for the same individual at different time points are linked by a black line. The size of the points represents the CN score at diagnosis (with black edges corresponding to scores ≤ 20) and the opacity represents the mean variant allele frequency (mVAF). Individual 8 was sampled twice on the same day. Asterisks indicate consensus genomes with <80% completeness.

## Viral haplotypes and clinical outcomes of SARS-CoV-2

At the beginning of the first wave of the pandemic, reports of asymptomatic infection or transmission were only beginning to emerge. Given that ~15% of individuals were asymptomatic or presymptomatic at the initial diagnosis, we sought to query if the presentation of symptoms (or lack thereof) was preferentially associated with a given viral genotype–a hypothesis that would significantly impact clinical management. The resolved viral haplotypes were largely not associated with clinical outcomes, as confirmed by other studies pertaining to the main viral subclades from the first wave [27, 28]. Medical predispositions and even environmental factors and geographical regions were shown to be important risk factors of severe and deadly cases of SARS-CoV-2 [29, 30]. Notwithstanding, our results suggest that clinical presentations are the predominant prognostic factors to consider when stratifying risk in COVID-19, at least with respect to the viral subclades and patient cohort in question. Interestingly, we report that the presentation of flu-like symptoms (e.g. myalgia, sore throat, headaches, S6 and S7 Figs) appears to be associated with more favorable patient outcomes, potentially indicative of an efficient and protective immune response. Headaches were also reported to be associated with younger age, fewer comorbidities and reduced mortality in a cohort of 379 Spanish COVID-19 patients from March 2020 [31]. Advanced age and the presence of comorbidities were, unsurprisingly, the main correlates of morbidity. The observed correlation between the CN score

(viral RNA abundance in nasopharyngeal sample) and the Charlson comorbidity index is counter-intuitive. Possible explanations are the low sample size (55 patients with an index >0) or less productive viral shedding in the nasopharynx in these patients. A less likely explanation could be that these patients present higher levels of anti-SARS-CoV-2 antibodies, which might inhibit viral replication (and, thus, increase the observed CN score) while potentially aggravating their condition through antibody-dependent enhancement [32]. Further serological studies in a larger cohort would be required to confirm this statement.

## Genomic diversity of SARS-CoV-2 in an early outbreak

The phylogenetic diversity of SARS-CoV-2 genomes we report in this local outbreak is consistent with the genomic diversity observed across Quebec at the time, specifically subclades B and B.1 [33]. However, we found that different lineage assignment methods we employed (Pangolin, NextClade, phylogeny, dominant haplogroups and PHATE dimensionality reduction) produced disparate results. Pangolin subclades localized to divergent branches of the maximum-likelihood tree we generated from multiple sequence alignments. The latter may be prone to grouping sequences with common amplicon dropouts, a common occurrence in our data, as we sequenced all SARS-CoV-2 positive samples. This composes a unique feature of our study, as current sequencing endeavours ignore samples with lower viral mRNA abundance (i.e. CN >20 or Ct >30). Despite the relatively low abundance of viral RNA in some of the samples, the majority of the consensus genomes we report are over 90% complete. The discrepancies that were observed between the phylogenic analysis and Pangolin classification are very unlikely to be associated with sequencing errors. Even though single-molecule sequencing is generally associated with a high single-read error rate (4–5% in the case of our data), the ARTIC pipeline has integrated error correction methods, based on banded event alignment or deep neural-networks. The fact that the default ARTIC parameter for the number of filtered reads was changed from 200 to 2000 per strand (see Material and Methods section) for our analysis ensures superior consensus accuracy. We can therefore assume that the majority of observed variants are *bona fide* mutations even though sequencing artifacts could be present in the data regardless of the sequencing technology used.

In April 2020, no publicly available SARS-CoV-2 genomes were available for Quebec. Given the rapid turnaround time of nanopore sequencing, we were able to upload the first 5 SARS-CoV-2 genomes from Quebec to GISAID [18] within a week of receiving the samples from Verdun hospital in a newly established laboratory with limited equipment and reagents. We anticipated that profiling the viral genomic diversity could be a useful epidemiology tool, potentially identifying specific transmission events and targeting specific measures to reduce hospital-acquired infections. However, this proved to be difficult given the relatively low genomic diversity observed in these 'first wave' samples and the short timespan between diagnoses. Nonetheless, the relatively higher frequency of the B.1/Haplogroup III in Verdun Hospital versus Notre-Dame Hospital indeed supports the suspected hospital-acquired infections and nosocomial transmission in this establishment. This is also substantiated by the significant enrichment of haplogroup III in patients with comorbidities, suggesting that this viral subclade may have been preferentially transmitted among hospitalized patients potentially sharing the same room or wing of the hospital. Indeed, this was to be anticipated given that the systematic use of personal protective equipment by both employees and patients was not mandated early in the COVID-19 pandemic. This also supports the observed presence of 'chimeric' genomes in our cohort, which may represent dual infection events caused by 2 different viral haplotypes, although we cannot rule out the unlikely possibility that this is a consequence of cross-contamination in these samples, despite the use of negative controls.

In contrast to Pangolin lineage assignment, we found that an *ad hoc* haplotype grouping strategy was globally consistent with the phylogenetic analysis and PHATE clusters. However, both Pangolin and the haplotype grouping failed to identify a significantly divergent subclade of SARS-CoV-2 in this outbreak (IIa/IIb), supporting the use of unsupervised methods for genomic sequence analysis (when possible). Indeed, when considering these additional subclades, we were able to identify a statistically significant association between subclade IIa and the presence of comorbidities and the presentation of headaches, albeit the latter may be a confounding effect of small sample size rather than a consequence of viral evolution itself. Recently, Mostefai et al. successfully used this method on all 2020 SARS-CoV-2 genomes available in GISAID, suggesting it can be used on datasets other than the one presented in this paper, containing more genomes and/or lineages [34].

Recent improvements to SARS-CoV-2 whole genome sequencing by tiled PCR amplicons, including the availability of ligation-based molecular barcodes for 96 samples, and to lineage assignment [12] have greatly facilitated the reliable analysis of SARS-CoV-2 genomes using nanopore sequencing [35]. The experimental and bioinformatics standard operating procedures developed by the ARTIC Network for SARS-CoV-2 [11] have been essential for the rapid and cost-effective sequencing of SARS-CoV-2, which can be performed from start to finish in less than 24h. These standard operating procedures were developed and optimized taking into consideration the best practices or analysis of SARS-CoV-2 that were discussed and published since the beginning of the pandemic [35–39].

We found that the medaka version of the ARTIC bioinformatics standard operating procedure provided consensus genomes with less ambiguous bases, as reported by others. However, we believe several improvements can be made to the standard bioinformatics analysis of SARS-CoV-2 nanopore sequencing data. Firstly, much of the ARTIC computational pipeline is single-threaded and could be parallelized to accelerate the bioinformatics turnaround time (particularly for smaller, independent labs with limited computing facilities). Secondly, we believe the default parameters may be a source of technical variation. We found that the quality filtering steps are robust, but the consensus generation steps––which assume a single subclade/haplotype is present––may be a source of artefacts, particularly when confronted to infections with more than one viral subclade or high CN/Ct scores (although many of these issues have been resolved by the global community since the commencement of this study). These might include 'chimeric' genomes where amplicon segments might alternate between predominant haplotypes in the consensus, which can explain some of the genomic outliers observed in this study (c.f. Fig 4, top row).

In conclusion, we posit that the rapid nanopore sequencing protocols for SARS-CoV-2, the accessibility of the platform, its low cost and ease of use are significant arguments for the widespread use of this platform for genomic epidemiology in local and global outbreaks. The rise of SARS-CoV-2 variants of concern around the globe combined with the progressive easing of public health restrictions and mass vaccination programs are additional reasons to implement rapid, lightweight genomic surveillance protocols, ideally directly at the point of care or diagnostic laboratory. For instance, performing nanopore sequencing of SARS-CoV-2 samples directly after positive diagnosis in a decentralized manner would (i) save on costs associated with RNA extraction and reverse-transcription, as SARS-CoV-2 sequencing requires the same material as diagnosis by quantitative PCR; (ii) identify genomic variants with a turnaround time of 24-48h; (iii) potentially provide recommendations for clinical management based on the identified variants; and (iv) facilitate effective, variant-focused contact-tracing measures during follow-up with the patient, which is impractical if not impossible to achieve with centralized, off-site next generation sequencing facilities.

## Materials and methods

### Diagnosis, sample collection and RNA extraction

Consecutive positive samples of SARS-CoV-2 collected from March 30th to April 17th, 2020, were provided by the clinical microbiology laboratory of Verdun Hospital. Those samples were collected from hospitalized patients, patients seen in the emergency department, and healthcare workers. The laboratory also processed samples from the associated Notre-Dame Hospital, a 250-bed general hospital located 10.3 km from Verdun Hospital, and those were included in the analysis.

Standard nasopharyngeal (NP) swabs were collected from patients and suspended in an RNA preservation and lysis solution. RNA extraction was then performed using TRIzol LS following the manufacturer's protocol (Invitrogen, California, United States of America). The presence of SARS-CoV-2 was detected using the Abbott RealTime SARS-CoV-2 assay (Abbott, Chicago, Illinois, United States of America) on an Abbott RealTime M2000rt, a qualitative multiplex real time PCR device that has FDA emergency use authorization for *in vitro* diagnostic use. This qualitative test gives a CN value for positive samples that is reminiscent of quantitative PCR values. CN values were manually extracted from digitized copies of the M2000rt reports.

### Nanopore sequencing

For each SARS-CoV-2 positive sample, 250 µl aliquots of NP swab were collected and stored at -80˚C until RNA extraction. 750 µl of TRIzol LS (Invitrogen, California, United States of America) was added to 250 µl of NP solution and RNA extraction was performed according to the manufacturers' recommendations. Final RNA pellets were resuspended in 15µl of nuclease-free water (Life Technologies, California, United States of America). For samples with a CN > 20, two aliquots were used for extraction, pellets were suspended in 10 µl of nuclease-free water and then pooled. 11 µl of the RNA was used for subsequent reverse transcription using SuperScript IV reverse transcriptase (Life Technologies, California, United States of America) and random hexamer primers (Life Technologies, California, United States of America).

Library preparation was performed following the Arctic Network nCov19 sequencing protocol version 1 (dx.doi.org/10.17504/protocols.io.bbmuik6w) and using individual V3 PCR primers (Life Technologies, California, United States of America) for samples included in the first sequencing run (e.g. "verd1"). The number of PCR cycles was determined based on the CN values: 25 cycles for a CN ≤ 7; 35 cycles for a CN > 7. For samples with a CN > 20, RNA was extracted from two aliquots, and 35 PCR cycles were used. Library preparation for runs verd2 and verd3 was performed following the similar Oxford Nanopore Technologies (ONT) PCR tiling of COVID-19 virus protocol and using manually pooled, individual ARTIC V3 PCR primers. The rest of the libraries were obtained and barcoded following the Oxford Nanopore Technologies PCR tiling of COVID-19 virus protocol and using the ARTIC nCoV-2019 V3 Panel (10006788, Integrated DNA Technologies, Iowa, United States of America) and ONT Ligation Sequencing Kit (SQK-LSK109, ONT, Oxford, United Kingdom). Two minor changes were made to the protocol, based on comparison with the Arctic Network nCov19 sequencing protocol version 2 (https://dx.doi.org/10.17504/protocols.io.bdp7i5rn): (i) At the reverse transcription step, samples were incubated for 5 minutes at 25˚C before incubation at 42˚C and (ii) samples were incubated for 10 minutes at both 20˚C and 65˚C during the end-prep step. Each sample was barcoded using Native Barcoding Expansion 1–12 and 13–24 kits (EXP-NBD104 and EXP-NBD114, ONT, Oxford, United Kingdom) and sequencing

performed on FLO-MIN006 (R9.4.1) flow cells using MinION MK1b, MinION MK1c and GridION sequencers (ONT, United Kingdom). Detailed technical information on the sequencing runs is listed in S2 Table.

### Data processing and consensus sequence generation

The raw sequencing files (.fast5) were base called offline with the proprietary basecaller Guppy, version 4.4.1 (ONT, Oxford, United Kingdom) using the R9.4.1 high accuracy configuration file. Base called reads were demultiplexed using Guppy with parameters "—require_-barcodes_both_ends—arrangements_files barcode_arrs_nb12.cfg barcode_arrs_nb24.cfg". The resulting reads were then size selected between 400 and 700 nt in length and subjected to the ARTIC Network Bioinformatics protocol (https://artic.network/ncov-2019/ncov2019-bioinformatics-sop.html) to generate full-length consensus genomes. Both the "nanopolish" and "medaka" parameters were used with default parameters, with the sole exception of increasing the "—normalise" parameter from 200 to 2000 for both commands.

The medaka consensus genomes were further processed to replace ambiguous ('N') bases at positions with lower variant allele frequencies, a consequence of (overly) strict variant filtering in the Medaka pipeline. This was performed by merging the "pass" and "fail" intermediary.vcf files, and filtering variants using the following parameters: (i) supported by at least 20 reads; (ii) not located in masked-regions as determined by the "coverage_mask.txt" files produced by the ARTIC pipeline; and (iii) present at a frequency above 50%. Resulting variants were inserted in the consensus sequence. The associated scripts can be found at https://github.com/TheRealSmithLab/Verdun. The coverage mask files were used to calculate genome completeness and to retain sequences for subsequent phylogenetic analyses.

### Phylogenetics

Assembled SARS-CoV-2 genome sequences with 80% or more completeness and the Wuhan-Hu-1 isolate reference genome (Genbank reference MN908947.3) were submitted to a multiple sequence alignment with MAFFT v7.475 using parameters "—maxiterate 500". The resulting multiple sequence alignment was used to generate a maximum-likelihood phylogeny using MEGA X [20]. The evolutionary history was inferred by using the Maximum Likelihood method and Tamura-Nei model [40]. The tree with the highest log likelihood was retained. Initial tree(s) for the heuristic search were obtained automatically by applying Neighbor-Joining and BioNJ algorithms to a matrix of pairwise distances estimated using the Tamura-Nei model, and then selecting the topology with superior log likelihood value. The tree is drawn to scale, with branch lengths measured in the number of substitutions per site. This analysis involved 235 nucleotide sequences. Codon positions included were 1st+2nd+3rd+Noncoding. All positions with less than 90% site coverage were eliminated, i.e. fewer than 10% alignment gaps, missing data, and ambiguous bases were allowed at any position (partial deletion option).

### Lineage calling

Lineage classification was performed using the Phylogenetic Assignment of Named Global Outbreak LINeages (Pangolin) software package (version 2.1.10) proposed by [12], using default parameters, as well as using the Nextstrain: real-time tracking of pathogen evolution (SARS-CoV-2 pipeline:2021/09/15). Nextstrain was built with a multiple input build-config (no filtering) using our sequences and metadata along with GISAID sequences and metadata (on 2021/09/16) [13].

## Haplotype grouping

A subset of 20 positions were selected to define viral haplotypes that represent large groups of sequences defined based on the worldwide most common genetic variants (using GISAID consensus sequences as of January 14th 2021) [34]. The 20 nucleotide positions are: 241, 313, 1059, 1163, 3037, 7540, 8782, 14408, 14805, 16647, 18555, 22992, 23401, 23403, 25563, 26144, 28144, 28881, 28882, 28883. These 20 mutations were selected because they exceeded 10% in variant allele frequency in at least one of the months of the first wave of the pandemic (January to July 2020) in GISAID consensus sequences. Further details on these haplotypes can be found in Mostefai et al. 2021. The definition of haplotypes based on these 20 mutations found in samples from this study, and their corresponding NextStrain clades, is reported in S5 Table. We note that haplotype III and IX are grouped within the same Nexstrain clade, despite the fact that they differ at position 25563, and both differ from haplotype II (NextStrain clade 20C) at position 14408.

## PHATE embeddings

PHATE embeddings [22] were computed independently for viral variants (257 features) and symptoms (43 features). In both cases, the input data for a given patient consisted of a binary encoding, with a 1-value for the presence of a given variant or symptom, and a 0-value for the absence thereof. PHATE is then applied to find a low-dimensional representation that preserves the geometry of the high-dimensional samples. The overall structure of the embeddings was relatively robust to the choice of hyperparameters. We therefore used the default parameters for PHATE, except for the diffusion-time parameter $t$, which was set to 30 for both the genotype and clinical data to display cleaner branches. Similarly, we set the *knn* value of MELD [23] (i.e. the number of considered neighbors) to 5 to be consistent with the PHATE default and the *beta* parameter (i.e. the amount of smoothing to apply) to 20 to avoid oversmoothing. We used PHATE 1.0.4 and MELD 1.0.0. Source code is respectively available at https://github.com/KrishnaswamyLab/PHATE and https://github.com/KrishnaswamyLab/MELD.

## Clinical data extraction

Data from patient files were extracted and entered in a database using a standardized case report form by an experienced research assistant (RR) and cross checked in full by IP.

## Ethics statement

Human research ethics approval for this study (MP-21-2021-2938) was provided by the CHU Sainte-Justine Research Centre ethics committee (FWA00021692) designated by the Quebec provincial government. Due to the retrospective nature of the study, and the absence of risk for participants, the need for consent was waived by the ethics committee, both for inclusion in the study, and for access to medical records for the purpose of data extraction. All patient samples have been de-identified.

## Supporting information

**S1 Fig. Amplicon coverage for all samples.** Amplicon coverage for each sample, including the controls, was calculated using bedtools using a 90% overlap between the query and the target (ARTIC Network amplicon coordinates) as well as 80% overlap between the target and the query.
(EPS)

**S2 Fig. Nextstrain genomic epidemiology of SARS-CoV-2.** Time-resolved phylogenetic tree of all genomes reported in our cohort (red dots) in the context of subsequent viral evolution (subsampled genomes from Nexstrain) during the COVID-19 pandemic (as of September 2021) produced via the Nextstrain SARS-CoV-2 pipeline.
(EPS)

**S3 Fig. Technical replicates for a selection of samples.** Two separate nanopore sequencing library preparations (1 & 2) from the same PCR products and the corresponding merged data (merged) on the horizontal axis. Results generated from the Medaka version of the ARTIC Network bioinformatics SOP. Genotype data generated with modified consensus genomes that contain the most frequent variant ($> 50\%$) at any given position.
(EPS)

**S4 Fig. PHATE embedding of SARS-CoV-2 genomic variation.** Each point corresponds to one SARS-CoV-2 genome annotated with various clinical labels and symptoms. The 15 most frequent variants are annotated in the bottom panel. Sub-clade IIa (right cluster) is characterized by a number of variants differentiating it from sub-clade IIb (tip of the lower branch), see Fig 4, top left panel.
(EPS)

**S5 Fig. MELD relative likelihood estimates based on viral genomic variation. Likelihoods of various clinical labels and symptoms are displayed over the PHATE embedding of SARS-CoV-2 genomes (see S3 Fig).** The subclade IIa cluster (top right right) appears depleted in adverse outcomes (mortality and hospitalization) and enriched in flu-like symptoms (sore throat, fatigue and headache). Conversely, the sub-clade IIb region (tip of the lower-right cluster) is slightly enriched in hospitalized patients and shows a higher likelihood of DEG/Confusion. MELD likelihoods of the 15 most frequent variants are displayed over the PHATE embedding of the genomes in the bottom panels. It should be stressed that MELD computes a local relative likelihood. Some manifold regions may be relatively depleted in a specific variant compared to other regions even if said variant is frequent in absolute terms, particularly in the case of 23403A>G, 3037C>T and 14408C>T.
(EPS)

**S6 Fig. PHATE embedding of clinical features.** Each point corresponds to one patient sample annotated with various clinical labels and symptoms (N.B. some samples correspond to the same patient, see Longitudinal Sequencing section). The 15 most frequent variants are annotated in the bottom panel. Sub-clade IIa (right cluster) is characterized by a number of variants differentiating it from sub-clade IIb (tip of the lower branch), see Fig 4, top left panel.
(EPS)

**S7 Fig. MELD relative likelihood estimates based on clinical features.** Likelihood estimates of various clinical labels and symptoms displayed over the PHATE embedding of the clinical features (see S5 Fig). The likelihood gradients of adverse out- comes (mortality, hospitalization and breathing assistance) are well aligned with comorbidity and DEG/Confusion gradients. Moreover, adverse outcome likelihoods appear to be inversely correlated with employee status as well as a set of flu-like symptoms (sore throat, myalgia, fatigue and headache). The MELD relative likelihood estimates of the 15 most frequent variants displayed over the PHATE embedding of the symptoms in the bottom panel.
(EPS)

**S1 Table. Sample overview.**
(XLSX)

**S2 Table. Run statistics.**
(XLSX)

**S3 Table. Negative controls.**
(XLSX)

**S4 Table. Technical replicates.**
(XLSX)

**S5 Table. Haplotype features.**
(XLSX)

**S6 Table. Genotype-phenotype comparisons.**
(XLSX)

**S7 Table. Longitudinal samples.**
(XLSX)

## Acknowledgments

We would like to thank members of the Mila COVID19 Task Force for their camaraderie and valuable insight into integrative data analysis strategies during the pandemic. Unrestrained gratitude and appreciation are expressed for members of the ARTIC Network, GISAID, and Nextstrain for their efforts and invaluable contribution to SARS-CoV-2 genomics. Thanks are owed to Jared Simpson and John Tyson for discussions relating to the interpretation of nanopore sequencing data, to Ioannis Ragoussis and Sarah Reiling for assistance with control experiments, to François Fontaine and Valérie Villeneuve for acquisition of equipment and consumables, and to essential workers during the pandemic.

## Author Contributions

**Conceptualization:** Ivan Pavlov, Martin A. Smith.

**Data curation:** Ronald Racette, Ivan Pavlov, Martin A. Smith.

**Formal analysis:** Bastien Paré, Marieke Rozendaal, Sacha Morin, Léa Kaufmann, Raphaël Poujol, Fatima Mostefai, Jean-Christophe Grenier, Henry Xing, Julie G. Hussin, Guy Wolf, Ivan Pavlov, Martin A. Smith.

**Investigation:** Bastien Paré, Marieke Rozendaal.

**Methodology:** Bastien Paré, Marieke Rozendaal, Shawn M. Simpson, Julie G. Hussin, Guy Wolf, Ivan Pavlov, Martin A. Smith.

**Resources:** Miguelle Sanchez, Ariane Yechouron.

**Supervision:** Julie G. Hussin, Guy Wolf, Ivan Pavlov, Martin A. Smith.

**Visualization:** Sacha Morin, Martin A. Smith.

**Writing – original draft:** Bastien Paré, Marieke Rozendaal, Ivan Pavlov, Martin A. Smith.

**Writing – review & editing:** Bastien Paré, Marieke Rozendaal, Shawn M. Simpson, Ivan Pavlov, Martin A. Smith.

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
