## [Decision Letter · Decision Letter 0]

11 Aug 2021

PONE-D-21-20458

Genomic epidemiology and associated clinical outcomes of a SARS-CoV-2 outbreak in a general adult hospital in Quebec

PLOS ONE

Dear Dr. Smith,

Thank you for submitting your manuscript to PLOS ONE. After careful consideration, we feel that it has merit but does not fully meet PLOS ONE’s publication criteria as it currently stands. Therefore, we invite you to submit a revised version of the manuscript that addresses the points raised during the review process.

While your paper addresses an interesting question, the reviewers stated several concerns about your study and did not recommend publication in present form.  In particular, the rationale of the study needs to be strengthen and focused.  The authors used two algorithms to assess their data:  PHATE and Pangolin.  The rationale of using these two methods needs to be mentioned in the Introduction.  The presentation also need to be improved.  In addition, there were numerous issues identified where additional experimentation and documentation is needed.  Please see reviewers’ insightful comments below.  On a personal level, I also have several questions that need to be clarified (see specific comments).

Specific comments:

Line 37, change “…Quebec (Canada)…” To “…Quebec, Canada…”Line 45 – 49, separate this section into second paragraph.  Strengthen the rationale of the study, and explain why the authors endeavored for whole genome sequencing, why use two algorithms and the decision process of choosing these two algorithms.Line 62, comorbidities, any comorbidities or specific comorbidities?Line 80, do you have a submission ID for GISAID? If so, please list here.Line 181:  “…the upper right we more…” should this be “…the upper right were more…”Line 184 – 185, this is an interesting statement, are any other reports documented the similar finding?Line 192:  “…146 out 209 infections…” should be “…146 out of 209 infections…”Line 214, change 360-000 to 360,000 and 127-000 to 127,000Reference 10 & 12 need to be updated.Figures 1, 2 & 4, please flip the x-axis label.

We look forward to receiving your revised manuscript.

Kind regards,

Baochuan Lin, Ph.D.

Academic Editor

PLOS ONE

Journal Requirements:

2. We note that you are reporting an analysis of a microarray, next-generation sequencing, or deep sequencing data set. PLOS requires that authors comply with field-specific standards for preparation, recording, and deposition of data in repositories appropriate to their field. Please upload these data to a stable, public repository (such as ArrayExpress, Gene Expression Omnibus (GEO), DNA Data Bank of Japan (DDBJ), NCBI GenBank, NCBI Sequence Read Archive, or EMBL Nucleotide Sequence Database (ENA)). In your revised cover letter, please provide the relevant accession numbers that may be used to access these data. For a full list of recommended repositories, see http://journals.plos.org/plosone/s/data-availability#loc-omics or http://journals.plos.org/plosone/s/data-availability#loc-sequencing.

3. PLOS ONE does not permit references to unpublished data; therefore, we request that you either include the referenced data or remove the instances of "data not shown," "unpublished results," or similar.

Reviewers' comments:

Reviewer's Responses to Questions

**Comments to the Author**

1. Is the manuscript technically sound, and do the data support the conclusions?

Reviewer #1: Partly

Reviewer #2: Yes

2. Has the statistical analysis been performed appropriately and rigorously? 

Reviewer #1: Yes

Reviewer #2: Yes

3. Have the authors made all data underlying the findings in their manuscript fully available?

Reviewer #1: Yes

Reviewer #2: Yes

4. Is the manuscript presented in an intelligible fashion and written in standard English?

Reviewer #1: Yes

Reviewer #2: Yes

5. Review Comments to the Author

Reviewer #1: It is an interesting study trying to associate between virus genomic and clinical outcomes of SARS-CoV-2 in Quebec, Canada. My suggestions are as follows:

- General comments: authors should choose between the association between virus genomic and clinical outcomes or the genomic epidemiology of SARS-CoV-2 in Quebec, to make easy for readers to understand the manuscript 's messages.

- Abstract: I have difficulty to understand the flow of abstract, please revise the abstract to reflect the story of manuscript

- Introduction: authors should emphasize the impact of study for the current knowledge of SARS-CoV-2 genomic since the samples were collected before introducing the VOI or VOC

- Results: please make subheadings to be more appropriate with the findings, for example: please revise the following subheadings: Nanopore sequencing of SARS-CoV-2 genomes  it is more suitable for subheadings of Methods section.

Figure 1 is more appropriate for suppl. Fig

It's better to provide Tables of association between lineage and outcomes, rather than using figures.

- Discussion: please make subheadings. please re-write the Discussion focusing on the implications of main findings.

Reviewer #2: Review of the paper "Genomic epidemiology and associated clinical outcomes of a SARS-CoV-2 outbreak in a general adult hospital in Quebec"

The paper characterizes the SARS-Cov-2 virus genomes sequenced last year between March-April 2020. The paper includes the clinical features of the samples and the authors tried to correlate these

clinical features to the virus genotypes. The authors addressed some issues related to the use of nanopore sequencing technology.

General Comments:

- The paper addresses the viral changes within the first wave spread last year 2020. It is not so late to publish these results, especially that the authors conducted the genome sequencing few days after the sample collection, as mentioned by the authors !!!. Nowadays one talks of 4th wave and new variants of the virus which further evolved beyond the original B, B.1, B.1.147 lineages. This is in my view is a major drawback, but it can be overcome if the authors would include some recent new sequences from 2021 and analyze them along with in-house sequenced ones; it is fine if they add sequences from Quebec or nearby areas deposited in GISAID.

- The paper does not include sufficient literature review either in the discussion or in the methodology. Best practices for analysis of SARS-Cov-2 using different platforms have been discussed in many papers since the emergence of the first sequences. Also the medical discussion about the association of the clinical features to haplotypes and the related mutations is not well enriched with references.

Specific Comments:

- The threshold of 80% coverage for accepting/rejecting sequences and using this for analysis is very low compared to usual practice of 90% at 10X and 95% at 1X. It is important to assure that the S and N genes do not have missing segments in this analysis.

- Did the author run pangoling in house or used the pangoling classification already in nextstrain?

- The section in Page 7 about discrepancies between phylogeny and pangloin gives the impression that pangolin generally failed on this issue. This issue needs careful discussion as a number of factors should be considered: Phylogeny algorithms favors more common variations in the clustering of samples due to the scoring system, and pangolin might be more sensitive for that in case of outliers or sequencing erros. One could test this by generating fasta files for the virus with the 20+ mutations only and presenting this to pangolin to compare pangolin to the phylogeny-based method or the in-house developed methods. Another dimension is that sequencing errors can dramatically affect pangolin performance, so more careful variant calling is important.

- Handling sequencing errors and variations (mutations) calling: ONT technology is known to have high rate of sequencing errors and many ambiguous mutation. The authors did not discuss any previous work related to handling this issue and no mention/reference of any best practices. The solution suggested by the authors, if it is novel, could have been supported by sequencing some samples using different method (e.g. Illumina or Sanger) and measuring the sensitivity/specificity of detecting the variations. [An example of best practice is to ignore mutations that appear once in own dataset and never shown up in world dataset.]

- The part of the paper related to identifying haplotyps or clusters is interesting. However, the authors referred to the methodology and this in turn referred to unpublished work. This part was presented as one of the contribution but nothing mentioned about it. In fact, there are clade assignment methodologies other than pangoling such as Cov-Glue and NextClade. Also the methodology where the author's method depends on known (high freq.) variantions needs to be more defended in case one introduced only these variations were introduced to the other lineage systems. I think including more description of this method and comparison to other known techniques is important.

- Genotype-Phenotype (clinical feature and genotype analysis) was performed on the haplotype level. One could also do this on individual variations and link this to effects on protein structure of certain genes. This would give more insight about this.

- The discussion section need also to be enriched with previous work linking clinical features to genomic variations and haplotypes. Which findings are considered novel and which ones are well known.

- It is better to describe the virus mutations using amino acids, in addition to physical coordinates,

to make it easier for the reader to follow. For example, the position A23403G is well known as the famous D614G mutation.

- Please define “co-morbidity”.

6. PLOS authors have the option to publish the peer review history of their article (what does this mean?). If published, this will include your full peer review and any attached files.

Reviewer #1: **Yes: **Gunadi

Reviewer #2: No

---

## [Author Response · Author response to Decision Letter 0]

20 Sep 2021

Editorial comments

 • In particular, the rationale of the study needs to be strengthen and focused. 

 • The authors used two algorithms to assess their data: PHATE and Pangolin. The rationale of using these two methods needs to be mentioned in the Introduction. 

 • The presentation also need to be improved. 

 • In addition, there were numerous issues identified where additional experimentation and documentation is needed. 

We have substantially revised the manuscript with an emphasis on clarifying the narrative. You will find a new title, enhanced abstract and introduction, as well as additions to the results and a more comprehensive discussion relating our findings to more recent ones reported in the literature. 

Specific comments:

 1. Line 37, change “…Quebec (Canada)…” To “…Quebec, Canada…”

Fixed. 

 2. Line 45 – 49, separate this section into second paragraph. Strengthen the rationale of the study, and explain why the authors endeavored for whole genome sequencing, why use two algorithms and the decision process of choosing these two algorithms.

We significantly expanded the introduction with additional information pertaining to the experimental and analytical tools described in the manuscript, as well as the motivation behind the study. 

 3. Line 62, comorbidities, any comorbidities or specific comorbidities?

Any comorbidities, as defined by a Charlson index > 0. We have clarified this in the revised manuscript. 

 4. Line 80, do you have a submission ID for GISAID? If so, please list here.

Each genome has a unique submission ID. They can be retrieved by searching the database for “Smith Laboratory” as the submitting lab, which we have clarified in the revised manuscript. 

 5. Line 181: “…the upper right we more…” should this be “…the upper right were more…”

 6. Line 184 – 185, this is an interesting statement, are any other reports documented the similar finding?

We have clarified this point in the revised manuscript.

 7. Line 192: “…146 out 209 infections…” should be “…146 out of 209 infections…”

 8. Line 214, change 360-000 to 360,000 and 127-000 to 127,000

 9. Reference 10 & 12 need to be updated.

Fixed all 3 points. 

 10. Figures 1, 2 & 4, please flip the x-axis label.

We are unsure what the issue with the labels is. In our hands, the labels are in the correct orientation. Could perhaps be a .eps formatting discrepancy? We used Adobe Illustrator to generate the .eps figures (they are saved as an Adobe illustrator EPS format). 

Reviewer 1

 • General comments: authors should choose between the association between virus genomic and clinical outcomes or the genomic epidemiology of SARS-CoV-2 in Quebec, to make easy for readers to understand the manuscript 's messages.

We appreciate and reflect the reviewer’s sentiment. The genomic epidemiology aspect is rather limited in its scope (revealing hospital-acquired transmission, which can be of utility to administrative staff and infectiologists). Therefore, we edited the manuscript to emphasize some of the caveats that we discovered with established analytic pipelines used for genotyping and genomic epidemiology, namely the poor discriminative ability of supervised lineage classification tools (i.e. Pangolin). The revised manuscript also showcases the strength of non-linear dimensionality reduction at identifying concrete relationships within complex multi-parametric data, both for genomic profiling and clinical presentations of infection. We have made the appropriate changes to reflect this pivot, namely by changing the title and reducing emphasis on epidemiology as the subject of this research. 

 • Abstract: I have difficulty to understand the flow of abstract, please revise the abstract to reflect the story of manuscript. 

In line with the other comments, we have amended the abstract to refine the narrative of the study. We trust these edits will emphasize the motivation and findings of the study. 

 • Introduction: authors should emphasize the impact of study for the current knowledge of SARS-CoV-2 genomic since the samples were collected before introducing the VOI or VOC

Indeed, our study was founded on some of the first infected patients in Canada, early in the ‘first wave’ of the pandemic. We observed viral genomic diversity, which was unknown at the time and was sufficient to validate hospital-based transmission. We report possible double-infection events and well-documented longitudinal samples, which are less common in public repositories, whilst providing a public resource for subsequent data mining of viral genomics and associated clinical response. Our study also shows that the observed viral genome diversity in this cohort had no impact on patient health outcomes–the main objective of this project–which can help guide similar genotype-phenotype association studies in the future. 

However, the main impact of our work arguably resides in our observation that viral genome diversity is not accurately depicted by popular lineage classification tools, which have since become established as a reference for viral subclade classification. We also compare different parameters for generating consensus sequences using one of the most commonly used pipelines for nanopore sequencing of SARS-CoV-2, exposing the inferiority of the default parameters. Our study thus demonstrates how routinely used tools in SARS-CoV-2 genomics can overlook substantial diversity in the underlying data, whereas lightweight, unsupervised methods (e.g. PHATE) offer an informative alternative for such applications. We believe the revised manuscript accentuates the impact of our study.

 • Results: please make subheadings to be more appropriate with the findings, for example: please revise the following subheadings: Nanopore sequencing of SARS-CoV-2 genomes  it is more suitable for subheadings of Methods section.

Thank you for this helpful suggestion. We have changed subheadings in the results to be more distinctive, as requested. Specifically:

“Clinical observations” to “Clinical observations and outcomes”;

“Nanopore sequencing of SARS-CoV-2 genomes” to “Viral RNA abundance and bioinformatics parameters affect genome assembly quality”;

“Phylogeny and lineage classification” to “Unsupervised machine learning outperforms supervised methods at discriminating between viral subclades ”;

 “Hospital-acquired transmission of SARS-CoV-2” to “Comparative sampling reveals hospital-acquired transmission of SARS-CoV-2”; 

“Longitudinal sequencing of SARS-CoV-2 positive subjects“ to “Longitudinal sequencing reveals possible double-infection events” 

We also added/split a section of the results pertaining to clinical data association, which we entitled: “Dimensionality reduction reveals associations between viral subclades, clinical features and patient outcomes”. 

 • Figure 1 is more appropriate for suppl. Fig

As we claim that certain reference methods in SARS-CoV-2 genomics may be problematic, we think that illustrating the distribution of technical results justifies the inclusion of Figure 1 in the main text. Furthermore, we believe it is important to show the relationship between CN and genome completeness, as there are few studies reporting CN values, despite the common use of the Abbott RealTime M2000rt device in SARS-CoV-2 diagnostics. 

 • It's better to provide Tables of association between lineage and outcomes, rather than using figures.

We provide both (c.f. Supplementary Table 1 and Figure 3). We argue that including the figure, albeit rich in information, enables a direct association between phylogeny and clinical outcomes, which would be difficult to assess in table format. The PHATE embeddings intrinsically provide an even more simplistic 2D representation of these relationships, although many readers familiar with phylogenetics may not appreciate this distinction. 

 • Discussion: please make subheadings. Please re-write the Discussion focusing on the implications of the main findings.

The discussion has been reorganized and enhanced accordingly. 

Reviewer 2

 • The paper addresses the viral changes within the first wave spread last year 2020. It is not so late to publish these results, especially that the authors conducted the genome sequencing few days after the sample collection, as mentioned by the authors !!!. Nowadays one talks of 4th wave and new variants of the virus which further evolved beyond the original B, B.1, B.1.147 lineages. This is in my view is a major drawback, but it can be overcome if the authors would include some recent new sequences from 2021 and analyze them along with in-house sequenced ones; it is fine if they add sequences from Quebec or nearby areas deposited in GISAID.

The principle objective of the study was to qualify viral diversity in one of the first outbreaks in Canada and to contrast this with clinical symptoms. We have thus reduced the emphasis on genomic epidemiology in the revised manuscript, while promoting the analytical aspects (i.e. the utility of unsupervised machine learning). 

We believe that the manuscript is scientifically valid, presents a strong methodology and high ethical standards, the fundamental publication criteria for PLoS ONE. We do not believe including more recent genomes would improve these qualities nor impact the conclusions of our study. However, we appreciate that the reviewer’s interest in placing the surveyed SARS-CoV-2 genomes in the context of current viral phylogenetics may reflect that of the journal’s readership. We have therefore included a supplementary figure that illustrates the time-resolved phylogenetic relationship between the genomes reported in this study and those subsequently sequenced from across the world (from NextStrain), as well as a sentence describing this at the end of the first paragraph of the “Unsupervised machine learning outperforms supervised methods at discriminating between viral subclades” section. 

 • The paper does not include sufficient literature review either in the discussion or in the methodology. Best practices for analysis of SARS-Cov-2 using different platforms have been discussed in many papers since the emergence of the first sequences. 

The genomes we report should be considered as some of these “first sequences”, as they were generated in early 2020, when few publications reported best practices for Oxford nanopore sequencing and data analysis. Our study implemented the most established methods for SARS-CoV-2 sequencing and variant calling at the time: The seminal ARTIC Network standard operating protocols, which remain the predominant protocol used for Oxford Nanopore sequencing of SARS-CoV-2 and is updated periodically based on community feedback, including our own. Although these were not published in a journal at the time, we have enhanced the revised manuscript with these references and several others, in line with the other reviewer comments. 

 • Also the medical discussion about the association of the clinical features to haplotypes and the related mutations is not well enriched with references.

We have added a section to the discussion which describes key reports pertaining to viral genotypes and patient phenotypes. Indeed, there weren’t many published reports about first-wave viral diversity and it’s impact on clinical outcomes, with the exception of the D614G mutation, which has been largely shown to increase virus fitness, but not symptoms or clinical outcomes. One reason for this is that there were (and still are) few publicly available datasets with detailed clinical features and associated viral genomes from infected individuals. In this regard, we posit that the public dissemination of both these data in our study will facilitate future data mining endeavours, as well as supporting the conclusions of our work. 

 • The threshold of 80% coverage for accepting/rejecting sequences and using this for analysis is very low compared to usual practice of 90% at 10X and 95% at 1X. It is important to assure that the S and N genes do not have missing segments in this analysis.

We are unfamiliar with the “usual practice of 90% at 10X and 95% at 1X” but would be open to the reviewer clarifying this statement with specific references. The 80% threshold was subjectively used as a minimum threshold for phylogeny and haplotype analysis to include as much of the data as possible. The sequences with [>80%,<90%] completeness represent a minority of the sequences (30/237, c.f. Figure1); the median genome completeness was 98%. Moreover, the phylogenetic inference parameters that ignore positions with more than 10% gaps were employed. We also did not observe enrichment for specific ‘genome gaps’ in the phylogenetic and PHATE clusters (not shown), therefore suggesting that the rare inclusion of genomes with up to 20% missing data had negligible–if any–influence on the lineage assignment and phylogenetic results. 

We would like to thank the reviewer for emphasizing the importance of inspecting the integrity of S and N genes. Following the reviewer’s comment, we noticed that 42/237 genomes had an incomplete S gene, and 90/237 had an incomplete N gene despite the use of conservative, best-practice analytic parameters. As a note, one of the consistently less abundant amplicons from the ARTIC V3 PCR amplification scheme occurs in the N gene (c.f. Supp Figure 1). However, only 4 unique mutations (7 mutations in total) were observed in 147 genomes with complete N genes, suggesting that the missing sequences are unlikely to contain many mutations. Few mutations were also observed for the S gene; besides the D614G mutation (present in all but one genome), 9 genomes had an A24782G mutation (N1074D a.a. substitution) and 7 had a G21641T (A27S a.a. substitution). The former was uniquely present in haplogroup IIa genomes, whereas the latter was found in genomes with haplogroup IIb classification 5 times and twice in haplogroup III. Since we observed significant enrichment for gastro-intestinal symptoms in haplogroup IIb and headaches in haplogroup IIa, we investigated if there was any significant association between these mutations and clinical features but found that the only significant association with between A24782G and headaches. However, both mutations are rarely observed in viral genomes sampled after our study (based on their representation in NextStrain). These findings have been merged into the results of the revised manuscript. 

 • Did the author run pangoling in house or used the pangoling classification already in nextstrain?

As stated in the Methods section: “Lineage classification was performed using the Phylogenetic Assignment of Named Global Outbreak LINeages (Pangolin) software package (version 2.1.10) proposed by (Rambaut et al. 2020), using default parameters.” 

The Pangolin lineage classifier was developed to enable dynamic, consistent naming, therefore a local install would provide the same classification IDs as those in public repositories. 

 • The section in Page 7 about discrepancies between phylogeny and pangloin gives the impression that pangolin generally failed on this issue. This issue needs careful discussion as a number of factors should be considered: Phylogeny algorithms favors more common variations in the clustering of samples due to the scoring system, and pangolin might be more sensitive for that in case of outliers or sequencing erros. One could test this by generating fasta files for the virus with the 20+ mutations only and presenting this to pangolin to compare pangolin to the phylogeny-based method or the in-house developed methods. Another dimension is that sequencing errors can dramatically affect pangolin performance, so more careful variant calling is important.

It is highly unlikely that sequencing errors contribute to the observed discrepancies seen in the phylogenetic analysis and Pangolin classification. Albeit single-molecule sequencing is associated with a high single-read error rate (4-5% in the case of our data), the ARTIC bioinformatics pipeline employs consensus-based error correction methods, based on either adapted banded event alignment (raw signal ‘polishing’ with nanopolish) or deep neural-networks (medaka). As stated in the methods, we modified the default ARTIC parameter for the number of filtered reads from 200 to 2000 per strand to ensure superior consensus accuracy. With 150 reads, the medaka error-correction tool outputs consensus sequences with Q40 (99.99%) accuracy (https://nanoporetech.github.io/medaka/benchmarks.html); we use up to >10x more data for consensus calling. A minimum threshold of 40 reads per amplicon is also applied, which is associated with a 99.95% consensus accuracy. We therefore can assume that the majority of observed variants are bona fide mutations, although sequencing artifacts could nonetheless be present, regardless of the sequencing technology. Moreover, non-random (consistent) errors would be observed in all samples, which was not the case for the considered genomic variants. We have added a section to the discussion to specify these points.

We are unsure what the review implies when mentioning that “pangolin might be more sensitive for that [favoring more common variations in the clustering of samples?] in case of outliers or sequencing erros”. The discrepancies between pangolin/haplogroup/nexclade and phylogeny/PHATE SARS-CoV-2 subclade assignment are one of the key results of this study. The former employ pre-defined mutation signatures to perform classification, which by design will ignore certain mutations that may not be prevalent in the reference datasets. As we show, these methods are therefore less sensitive to genomic variation than phylogeny or unsupervised clustering techniques (such as enabled by PHATE). 

We respectfully disagree that sequencing errors would “dramatically” affect pangolin classification, as for this to happen, errors would have to perfectly overlap several positions identified as discriminative features by the pangolearn algorithm–an unlikely (albeit not impossible) outcome given the size of the SARS-CoV-2 genome (~30k bases), the estimated consensus error rate (≤0.01%) and the amount of discriminative positions on the pangolin decision tree (varies by subclade, from ~6 to >20). 

 • Handling sequencing errors and variations (mutations) calling: ONT technology is known to have high rate of sequencing errors and many ambiguous mutation. The authors did not discuss any previous work related to handling this issue and no mention/reference of any best practices. The solution suggested by the authors, if it is novel, could have been supported by sequencing some samples using different method (e.g. Illumina or Sanger) and measuring the sensitivity/specificity of detecting the variations. [An example of best practice is to ignore mutations that appear once in own dataset and never shown up in world dataset.]

We would invite the reviewer to refer to previous comments about the error rate. Other high-impact publications have described the comparison of Nanopore to Illumina sequencing (Bull et al. 2020; Xiao et al. 2020), one of which was written by close collaborators that we referred to in the original submission (the other has been added to the revised manuscript). We believe that a technical comparison of sequencing technologies is beyond the scope of this study, particularly since this has been previously reported in the litterature. We also specifically referred to the ARTIC Network’s laboratory and bioinformatics standard operating procedure (best practice) for SARS-CoV-2 nanopore sequencing data generation and analysis. Finally, we respectfully disagree that ignoring new mutations is best practice but we appreciate that a single outlier mutation may correspond to an error and can be ignored. The mutations we observe, however, occur in several samples. 

 • The part of the paper related to identifying haplotyps or clusters is interesting. However, the authors referred to the methodology and this in turn referred to unpublished work. This part was presented as one of the contribution but nothing mentioned about it. 

Thank you for your interest in this aspect of our study. We appreciate the conflicting reference to unpublished work, which is part of a broader study currently under second round of revision in another journal. However, we would like to highlight that the pertinent aspects of this method are in fact clearly detailed in the methods under the “Haplotype grouping” section and Table S5. For convenience: “... The 20 nucleotide positions are: 241, 313, 1059, 1163, 3037, 7540, 8782, 14408, 14805, 16647, 18555, 22992, 23401, 23403, 25563, 26144, 28144, 28881, 28882, 28883. These 20 mutations were selected because they exceeded 10% in variant allele frequency in at least one of the months of the first wave of the pandemic (January to July 2020) in GISAID consensus sequences. ...”. The reviewer might appreciate that this methodology is very similar to the ones proposed by pangolin and nextclade, where a set of discriminative mutation features are used to categorize viral subclades (continued in the next response). 

 • In fact, there are clade assignment methodologies other than pangoling such as Cov-Glue and NextClade. Also the methodology where the author's method depends on known (high freq.) variantions needs to be more defended in case one introduced only these variations were introduced to the other lineage systems. I think including more description of this method and comparison to other known techniques is important.

We are unsure what the reviewer means when stating “in case one introduced only these variations were introduced to the other lineage systems”. The haplogroup method we describe is in essence highly similar to the NextClade method (established in September 2020; not available when we started the study). We have nonetheless added the NextClade assignments to Table S1 in the revised manuscript, next to Pangolin and Haplogroup classifications, as well as in the Methods section. To our knowledge, CovGlue employs a webapp that annotates specific amino acid substitutions–it does not perform lineage classification/clade assignment in an original manner but reports amino acid changes and potential incompatibility with commonly used assays. CovGlue uses GISAID lineage nomenclature, which now includes Pangolin. We therefore did not include CovGlue in the revised manuscript. 

 • Genotype-Phenotype (clinical feature and genotype analysis) was performed on the haplotype level. One could also do this on individual variations and link this to effects on protein structure of certain genes. This would give more insight about this.

We have done this for the 15 most common variants, as shown in Supp Figures 4-6. Given that there was no statistically significant enrichment of given haplotypes in the surveyed clinical features (beyond what could be attributed to a sampling bias, which we discussed in the original submission) pursuing the functional impact of specific mutations was not deemed worthwhile for the purpose of this study. However, the data are publicly available for others to investigate this effect. 

 • The discussion section need also to be enriched with previous work linking clinical features to genomic variations and haplotypes. Which findings are considered novel and which ones are well known.

We have enhanced the discussion in the revised manuscript following the reviewer’s suggestion, including several recent references discussing the impact of genomic variation on clinical features that may not have been available when first drafting the manuscript. 

 • It is better to describe the virus mutations using amino acids, in addition to physical coordinates to make it easier for the reader to follow. For example, the position A23403G is well known as the famous D614G mutation.

We describe specific genomic variants at line 166 of the original manuscript, where we use both genomic and protein coordinates (e.g. 4886C>T, ORF1a.P1541S) using Human Genome Variation Society nomenclature standards for the unique variants reported in our study. In the supplementary data, we also describe the physical coordinates in genomic coordinates for compatibility with Variant Call Files (.vcf) and thus to facilitate lookup in the raw data. Moreover, following the reviewers interest in describing amino acid variants, we have uploaded the metadata and .json configuration files associated with the NextStrain/Auspice visualization we report in the Supplementary Figure 2 to the study’s github repository so that users can explore these data interactively using the NextStrain software packages (alternatively, the fasta sequences can be directly uploaded to the NextClade webportal). 

 • Please define “co-morbidity”.

Co-morbidity is a standard medical term that describes the presence of one or more additional conditions often co-occurring with a primary condition. In this manuscript, co-morbidities are represented by the Charlson comorbidity index, a quantitative metric premised on clinical features and developed to predict the ten-year mortality for a patient who may have a range of comorbid conditions. We have amended the manuscript to include these definitions.

---

## [Decision Letter · Decision Letter 1]

26 Oct 2021

PONE-D-21-20458R1 Patient health records and whole viral genomes from an early SARS-CoV-2 outbreak in a Quebec hospital reveal features associated with favorable outcomes

PLOS ONE

Dear Dr. Smith,

Thank you for submitting your manuscript to PLOS ONE. After careful consideration, we feel that it has merit but does not fully meet PLOS ONE’s publication criteria as it currently stands. Therefore, we invite you to submit a revised version of the manuscript that addresses the points raised during the review process.

One of the reviewers still has some issues with the sequencing parameters, limitation and clarification on the efforts in inspecting the S & N genes.  Please see reviewer's insightful comments below.   

We look forward to receiving your revised manuscript.

Kind regards,

Baochuan Lin, Ph.D.

Academic Editor

PLOS ONE

Reviewers' comments:

Reviewer's Responses to Questions

**Comments to the Author**

1. If the authors have adequately addressed your comments raised in a previous round of review and you feel that this manuscript is now acceptable for publication, you may indicate that here to bypass the “Comments to the Author” section, enter your conflict of interest statement in the “Confidential to Editor” section, and submit your "Accept" recommendation.

Reviewer #1: All comments have been addressed

Reviewer #2: All comments have been addressed

2. Is the manuscript technically sound, and do the data support the conclusions?

Reviewer #1: Yes

Reviewer #2: Partly

3. Has the statistical analysis been performed appropriately and rigorously? 

Reviewer #1: Yes

Reviewer #2: N/A

4. Have the authors made all data underlying the findings in their manuscript fully available?

Reviewer #1: Yes

Reviewer #2: Yes

5. Is the manuscript presented in an intelligible fashion and written in standard English?

Reviewer #1: Yes

Reviewer #2: Yes

6. Review Comments to the Author

Reviewer #1: Authors have addressed all comments appropriately. Thank you for the opportunity to review your work. Congratulation!

Reviewer #2: Review of the paper "Patient health records and whole viral genomes from an early SARS-CoV-2 outbreak in a Quebec hospital reveal features associated with favorable outcomes"

General Comments:

The authors exerted good effort to address my comments. There are few points that still need to be improved. The authors have chosen to focus more in the revised version on the methodology and clinical associations. This is in my view good choice especially the sequencing data of the paper represents older wave of the viral evolution.

Specific Comments:

- The authors need to comment on the limitations of the sequencing that some positions were not covered by the genome and the authors guessed them using dominant VAF? How many of these variant existed and what is the number of the genomes affected by this?? How this would affect the results if there were wrong predictions.??

- The authors need to mention their effort in inspecting the S and N genes and how the missing positions do not affect much their conclusion as they mentioned in the response to my revision. I missed this part in the results section.

- For best practice, the user should mention the sequencing parameters (depth and coverage) in their dataset and compare this to the usually obtained results as per the paper of Bull et al paper. Bull et al. mentioned coverage of 99.6% and tested sensitivity down to 50X read depth: They stated that sensitivity and precision of variant detection were strongly influenced by sequencing coverage, showing a sharp decline below ~50-fold coverage depth. (This can mean that one can target 99% coverage at 50X depth.) The authors need to comment on that and put the reader in context about the sequencing quality in this paper. How many of their sequences reached that level and why they still retained them in the analysis.

- The authors need to state that their method worked well compared to Pangolin/Nextstrain only for this specific data set, and this cannot be generalized to other data-sets due to the lack of more extensive experimentation using larger sequencing set and more lineages.

- I would drop marketing statements like price of nanopore (CA$50 per sample); or if it is crucial for some reason to mention that, then plz state the cost of other technologies as well, taking sequencing parameters (sequencing and depth) per sample to reach accepted level of accuracy into consideration.

7. PLOS authors have the option to publish the peer review history of their article (what does this mean?). If published, this will include your full peer review and any attached files.

Reviewer #1: **Yes: **Gunadi

Reviewer #2: No

---

## [Author Response · Author response to Decision Letter 1]

3 Nov 2021

Comments to the Author

Reviewer #1: Authors have addressed all comments appropriately. Thank you for the opportunity to review your work. Congratulation!

Thank you for your input. 

Reviewer #2: Review of the paper "Patient health records and whole viral genomes from an early SARS-CoV-2 outbreak in a Quebec hospital reveal features associated with favorable outcomes"

General Comments:

The authors exerted good effort to address my comments. There are few points that still need to be improved. The authors have chosen to focus more in the revised version on the methodology and clinical associations. This is in my view good choice especially the sequencing data of the paper represents older wave of the viral evolution.

Specific Comments:

- The authors need to comment on the limitations of the sequencing that some positions were not covered by the genome and the authors guessed them using dominant VAF? How many of these variant existed and what is the number of the genomes affected by this?? How this would affect the results if there were wrong predictions.??

The ARTIC Bioinformatics Standard Operating Procedure for nanopore requires that at least 20 reads per strand per amplicon are present to generate a consensus – we stuck to these parameters for consistency with other ARTIC-nanopore consensus sequences in GISAID. Regions not covered by at least 20 reads on either strand are therefore not included in the consensus (NNNNN...). At no point do we ‘guess’ a variant. The reviewer may be referring to situations when an allele frequency below ~0.9 is observed, which causes the default variant calling pipeline to occasionally emit ambiguous “N” variants instead. In these situations, which are independent of coverage, we extracted the major allele in the consensus sequence, which is default behaviour for haploid genomes. This post-processing we performed corrects a recurring artifact in the ARTIC pipeline when using medaka, which has been reported by ARTIC pipeline developers (https://community.artic.network/t/medaka-longshot-pipeline/107). 

- The authors need to mention their effort in inspecting the S and N genes and how the missing positions do not affect much their conclusion as they mentioned in the response to my revision. I missed this part in the results section.

Thank you for this recommendation. We added the summary of this analysis (c.f. previous response to reviewers) on page 8 and 10-11 of the results, Specifically: 

Of note, 42/237 genomes with ≥80% completeness had an incomplete S gene and 90/237 had an incomplete N gene. The latter harbors one of the consistently less abundant amplicons from the ARTIC V3 PCR amplification scheme (S1 Fig). However, only 4 unique mutations (7 mutations in total) were observed in 147 genomes with complete N genes, suggesting that the missing sequences are unlikely to contain many mutations. Few mutations were also observed for the S gene; besides the D614G mutation (present in all but one genome), 9 genomes had an A24782G mutation (N1074D substitution) and 7 had a G21641T (A27S substitution).

…

All 9 A24782G mutations in the N gene (see above) were uniquely present in haplogroup IIa genomes, whereas 5/7 G21641T mutations in the S gene were present in haplogroup IIb genomes (the remaining 2 in haplogroup III).

- For best practice, the user should mention the sequencing parameters (depth and coverage) in their dataset and compare this to the usually obtained results as per the paper of Bull et al paper. Bull et al. mentioned coverage of 99.6% and tested sensitivity down to 50X read depth: They stated that sensitivity and precision of variant detection were strongly influenced by sequencing coverage, showing a sharp decline below ~50-fold coverage depth. (This can mean that one can target 99% coverage at 50X depth.) The authors need to comment on that and put the reader in context about the sequencing quality in this paper. How many of their sequences reached that level and why they still retained them in the analysis.

The depth and coverage from our dataset are illustrated in Table 1 as well as detailed in Supplementary Figure 1. As for comparing the depth and coverage we obtained to the ones presented by Bull et al., it is important to note that their methodology is distinct and was developed after we published the consensus genomes associated with this publication. Their protocol is optimised for nanopore sequencing (~2.5 kb-long amplicons), whereas the ARTIC V4 primer scheme we used was intended for compatibility across Illumina and Nanopore platforms, for broader community adoption. 

 Furthermore, the Bull et al. study and most SARS-CoV-2 sequencing protocols typically one process samples with higher viral mRNA quantities (Ct scores < ~30 or CN scores < ~20, depending on local recommendations). As we shown in Figure 1B, lower scores (more RNA) are associated with more even amplicon coverage and, consequently, higher genome completeness. As mentioned in the discussion, a unique feature of our study is that we included samples with lower RNA abundance (i.e. leading to low coverage), which complicates variant calling and lineage assignment, but can provide information on intra-host variation and viral evolutionary dynamics–two aspects not typically addressed in studies tacking genomic epidemiology alone.

 We would like to remind the reviewer that a minimum of 40-fold coverage (20 on both strands of an amplicon) are required to produce a consensus sequence with the ARTIC bioinformatics SOP. Given the dynamics of RT-PCR and the ARTIC V3 primer scheme, some amplicons have systematically higher or lower coverage than others (see comments in the previous response), which can cause coverage dropouts as illustrated in Supp Figure 1. Nevertheless, we obtained a median completeness of 97.7% with short amplicons (about 400 bp) for all genomes, with CN ranges between 0-31. This includes 70 genomes with full-coverage (99.6%), or 26.5% of our samples (a phrase was added to specify this in the results on page 6). In contrast, Bull et al. obtained 99.6%, or the maximum possible with ARTIC primer schemes, using longer amplicons and Ct ranges ≤ 29. N.B. a Ct score of 33 is approximately equivalent to a CN score of 22. 

 We believe that readers interested in the sequencing quality of our study will appreciate the detailed metrics provided in Figure 1, Supplementary Table 1, Supplementary Figure 1 and the summary in the body of the Results section, as well as recognizing the use of the internationally-implemented ARTIC standard sequencing and bioinformatics protocol used by hundreds of laboratories sequencing SARS-CoV-2. Therefore, we do not think this should be further emphasized. 

- The authors need to state that their method worked well compared to Pangolin/Nextstrain only for this specific data set, and this cannot be generalized to other data-sets due to the lack of more extensive experimentation using larger sequencing set and more lineages.

Recently, Mostefai et al., (Mostefai et al. 2021) also showed that this method can be used on larger sequencing sets (i.e. all 2020 SARS-CoV-2 genomes in GISAID), as well as more lineages. Thus, we believe this method can be used on datasets other than the one presented in this paper.

- I would drop marketing statements like price of nanopore (CA$50 per sample); or if it is crucial for some reason to mention that, then plz state the cost of other technologies as well, taking sequencing parameters (sequencing and depth) per sample to reach accepted level of accuracy into consideration.

We agree with the reviewer and the statements were removed in the updated manuscript.

---

## [Decision Letter · Decision Letter 2]

16 Nov 2021

Patient health records and whole viral genomes from an early SARS-CoV-2 outbreak in a Quebec hospital reveal features associated with favorable outcomes

PONE-D-21-20458R2

Dear Dr. Smith,

We’re pleased to inform you that your manuscript has been judged scientifically suitable for publication and will be formally accepted for publication once it meets all outstanding technical requirements.

Kind regards,

Baochuan Lin, Ph.D.

Academic Editor

PLOS ONE

Additional Editor Comments (optional):

Reviewers' comments:

Reviewer's Responses to Questions

**Comments to the Author**

1. If the authors have adequately addressed your comments raised in a previous round of review and you feel that this manuscript is now acceptable for publication, you may indicate that here to bypass the “Comments to the Author” section, enter your conflict of interest statement in the “Confidential to Editor” section, and submit your "Accept" recommendation.

Reviewer #2: All comments have been addressed

2. Is the manuscript technically sound, and do the data support the conclusions?

Reviewer #2: Yes

3. Has the statistical analysis been performed appropriately and rigorously? 

Reviewer #2: N/A

4. Have the authors made all data underlying the findings in their manuscript fully available?

Reviewer #2: Yes

5. Is the manuscript presented in an intelligible fashion and written in standard English?

Reviewer #2: Yes

6. Review Comments to the Author

Reviewer #2: The authors addressed all comments and the paper can be now accepted.

The authors addressed all comments and the paper can be now accepted.

7. PLOS authors have the option to publish the peer review history of their article (what does this mean?). If published, this will include your full peer review and any attached files.

Reviewer #2: No

---

## [Editor Report · Acceptance letter]

22 Nov 2021

PONE-D-21-20458R2 

Patient health records and whole viral genomes from an early SARS-CoV-2 outbreak in a Quebec hospital reveal features associated with favorable outcomes 

Dear Dr. Smith:

I'm pleased to inform you that your manuscript has been deemed suitable for publication in PLOS ONE. Congratulations! Your manuscript is now with our production department. 

Kind regards, 

on behalf of

Dr. Baochuan Lin 

Academic Editor

PLOS ONE